# Unique Metabolic Profiles of Korean Rice According to Polishing Degree, Variety, and Geo-Environmental Factors

**DOI:** 10.3390/foods10040711

**Published:** 2021-03-26

**Authors:** Yujin Kang, Bo Mi Lee, Eun Mi Lee, Chang-Ho Kim, Jeong-Ah Seo, Hyung-Kyoon Choi, Young-Suk Kim, Do Yup Lee

**Affiliations:** 1Center for Food and Bioconvergence, Research Institute for Agriculture and Life Sciences, Department of Agricultural Biotechnology, CALS, Seoul National University, Seoul 08826, Korea; yousin7716@naver.com (Y.K.); em2422@snu.ac.kr (E.M.L.); demizor@snu.ac.kr (C.-H.K.); 2BK21 Plus Program, Department of Bio and Fermentation Convergence Technology, Kookmin University, Seoul 02707, Korea; gook68@naver.com; 3School of Systems Biomedical Science, Soongsil University, Seoul 06978, Korea; sja815@ssu.ac.kr; 4College of Pharmacy, Chung-Ang University, Seoul 06911, Korea; hykychoi@cau.ac.kr; 5Department of Food Science and Engineering, Ewha Womans University, Seoul 03760, Korea

**Keywords:** brown rice, white rice, primary metabolites, secondary metabolites, variety, cultivation region, metabolomics

## Abstract

The precise determination of the chemical composition in crops is important to identify their nutritional and functional value. The current study performed a systematic delineation of the rice metabolome, an important staple in Asia, to investigate the following: (1) comparative features between brown and white rice; (2) variety-specific composition (Ilpum vs. Odae); and (3) cultivation of region-dependent metabolic content. Global metabolic profiling and data-driven statistics identified the exclusive enrichment of compounds in brown rice compared to white rice. Next, the authors investigated a variety-governed metabolic phenotype among various geo-environmental factors. Odae, the early-ripening cultivar, showed higher contents of most chemicals compared to the late-ripening cultivar, Ilpum. The authors identified regional specificity for cultivation among five areas in Korea which were characterized by polishing degree and cultivar type. Finally, the current study proposes a possible linkage of the region-specific metabolic signatures to soil texture and total rainfall. In addition, we found tryptophan metabolites that implied the potential for microbe–host interactions that may influence crop metabolites.

## 1. Introduction

Cereal grain is the third most widely produced agricultural product [1]. Oryza glaberrima or Sativa seeds are popularly grown in the African and Asian regions. Oryza Sativa has two major subspecies (japonica and indica) with a tremendous number of varieties. Japonica rice is the most widely cultivated rice in East Asia, particularly in Korea, China, and Japan [2].

In addition to caloric intake, the basal role of food, taste, and functional value are emerging interests among consumers and are mainly determined by the chemical composition. The compositional characteristics of food materials are affected by multiple complex factors. Similar to other agricultural products, the metabolite profiles of rice are dominated by genotype (variety) and geo-environmental characteristics (e.g., amount of rainfall, temperature, and soil) [3,4,5]. In particular, the metabolite contents are significantly different according to polishing processing (the transition of brown rice to white rice). Rice bran and germ are removed during the milling process. They contain various types of beneficial nutrients and are, thus, major contributors to the characterization of the metabolic profiles of rice.

Accordingly, five cultivation regions in Korea were chosen and paired with two Korean representative cultivars (Ilpum and Odae). The major difference between the varieties is their ripening period. Originally bred from the Ilpum cultivar, the cultivar Odae is an early-ripening cultivar which is mainly cultivated in Kangwon Province, a relatively cold area (northeastern area of Korea) [6]. Ilpum, a late-ripening cultivar, is ideal for cultivation regions with higher temperatures (e.g., Kyungsang Province, southern Korea).

The current study applied metabolomics as the major analytical approach. This method is an omics technology that is capable of directly tracing ‘chemicals’, and can thus comprehensively determine the nutritional characteristics of food. Although a few studies have conducted targeted or untargeted metabolite profiling of rice, this is the first to conduct a comprehensive metabolic evaluation of rice while simultaneously considering its major determinant factors. A recent study proposed that rice had discriminant lipid molecules, but was limited to its geographical origin (China and Korea) for discriminant model purposes [2]. Likewise, an earlier study reported discriminant metabolites of rice from different geographical origins in China based on nuclear magnetic resonance (NMR) spectroscopy [7]. Other studies showed the metabolic variation between japonica and indica using Liquid Chromatography-Mass Spectrometry (LC-MS) and Gas Chromatography-Mass Spectrometry (GC-MS) [8]. This study was conducted on 12 cultivars from India based on GC-MS [9], and on 17 cultivars from 11 countries using LC-MS [10]. Comparably, we explored the metabolic profiles of rice compartments (brown vs. white rice) and varieties (Odae vs. Ilpum) [6] coupled to five different cultivation regions in Korea. The results implied that the metabolites of rice seeds exhibit unique traits that are interactively determined by cultivar and cultivation region in brown and white rice.

## 2. Materials and Methods

### 2.1. Sample Collection

Rice samples were harvested in 2018 and collected by the National Institute of Crop Science. The samples included two varieties (Ilpum and Odae) with two polishing types (brown and white rice). These samples were cultivated in Chuncheon (Gangwon-do, Korea), Suwon (Gyeonggi-do), Cheongju (Chungcheongbuk-do), Sangju (Gyeongsangbuk-do), and Jeonju (Jeollabuk-do). The information is summarized in the Appendix A (Appendix A).

### 2.2. Metabolite Extraction

Rice grains (20 mg per sample) were lyophilized and pulverized using Mixer Mill MM400 (Retsch GmbH & Co., Haan, Germany). The powder was mixed with 1 mL of extraction solvent (methanol:isopropanol:water, 3:3:2, *v*/*v*/*v*) followed by sonication (5 min) and centrifugation (5 min, 16,100× *g* at 4 °C). The supernatant (400 μL for GC-TOF MS and 50 μL for Liquid Chromatography-Orbitrap Mass Spectrometry (LC-Orbitrap MS) was completely dried by a speed vacuum concentrator (SCANVAC, Lynge, Denmark).

### 2.3. Gas Chromatography Time-of-Flight Mass Spectrometry Analysis

For the first derivatization step, the dried extract was mixed and incubated with 5 µL of 40 mg/mL methoxyamine hydrochloride (Sigma-Aldrich, St. Louis, MO, USA) in pyridine (Thermo, Waltham, MA, USA) (90 min at 800 rpm at 30 °C). For the second derivatization, the mixture was reacted with 45 µL N-methyl-N-(trimethylsilyl)trifluoroacetamide (MSTFA + 1% TMCS; Thermo, USA) for 1 h [5,11]. Fatty acid methyl esters (mixture of 13 FAMEs, C8–C30) were added to the reactant as retention index markers. The derivative (0.5 µL) was injected with an Agilent 7693 Automatic Liquid Sampler (ALS) (Agilent Technologies, Wilmington, DE, USA) in splitless mode. The metabolites were chromatographically separated on an RTX-5Sil MS column (Restek, Gellefonte, PA, USA) while being controlled by an Agilent 7890B gas chromatograph (Agilent Technologies) [5,12]. Mass spectrometric data (85–500 m/z at 17 spectra s-1) were acquired by a Leco Pegasus High Throughput (HT) time of flight mass spectrometer (LECO Corporation, St. Joseph, MI, USA). The transfer line and ion source temperatures were set to 280 °C and 250 °C, respectively, with a detector voltage of 1850 eV (300 s solvent delay). For quality control purposes, a mixture of 33 compounds was analyzed every six samples.

Data preprocessing was conducted by ChromaTOF software v. 4.50 (LECO Corporation, St. Joseph, MI, USA). Then, the data were exported to our server computer for postprocessing, which included a data format conversion (text and NetCDF file) and evaluation (peak alignment, quality check, and quant ion selection). These analyses were based on the *Binbase* algorithm [13]. Peak height, based on a quant ion, was used for the quantitative value of each metabolite and was annotated against the Fiehn Library. Refer to our previous reports for details [5,12].

### 2.4. Liquid Chromatography-Orbitrap Mass Spectrometric Analysis

The dried extract was reconstituted with 50 µL of 80% MeOH. The reconstituent was chromatographically separated by a Waters Acquity Ultra Performance Liquid Chromatography (UPLC) BEH C18 column (2.1 mm × 150 mm, 1.7 μm) and an Ultimate-3000 UPLC system (Thermo Fisher Scientific, MA, USA). The mobile phase consisted of A (water with 0.1% formic acid, *v*/*v*) and B (acetonitrile with 0.1% formic acid, *v*/*v*). The gradient was set to the following: 0–2.0 min, 0% B; 2.0–30.0 min, 0–100% B; 30.0–32.0 min, 100% B; 32.0–32.1 min, 100–0% B; 32.1–35.0 min, 0% B. A Q-Exactive Plus instrument (Thermo Fisher Scientific, Waltham, MA, USA) was used in positive mode for mass spectrometric analysis. A full mass spectrum (MS) scan was conducted and ranged from 100–1500 Da (resolution of 70,000 FWHM), and MS/MS was performed in a data-dependent manner (High Energy Collision Dissociation (HCD): 30 eV, resolution of 17,500 FWHM). Data processing was performed using Compound Discoverer (version 3.1, Thermo Fisher Scientific, MA, USA). Peak alignment was performed within a mass tolerance of 5 ppm, and a retention time shift was allowed for 1 min. For compound identification, mass windows of 5 ppm and 10 ppm were applied for MS1 and MS2, respectively, with 70% similarity scores against the mzCloud library [5,12]. Pooled samples were injected every eight samples and further analyzed for quality control purposes.

### 2.5. Statistical Analysis

Statistical analyses were conducted on all continuous variables (metabolites). The data matrix from Gas Chromatography-Time of Flight Mass Spectrometry (GC-TOF MS) and LC-Orbitrap MS were log-transformed. Treemap was generated using Microsoft Excel (Microsoft, Seattle, WA, USA) based on compound classification (class and subclass) by the Human Metabolome Database (HMDB). Student’s *t*-test was performed using EXCEL (Microsoft Office 2016). Multivariate statistics, including principal component analysis (PCA) and orthogonal projection to latent structure-discriminant analysis (OPLS-DA), were performed using SIMCA 15 (Umetrics AB, Umea, Sweden). The OPLS-DA model was validated based on five-fold cross-validation. A chemical enrichment analysis was performed to evaluate statistical significance at the level of chemical class based on the ChemRICH program [5,14]. Box and whisker plots were generated using GraphPad Prism 7 (GraphPad Software Inc., San Diego, CA, USA). Percentages of variation of metabolite profiles were calculated by permutation multivariate analysis of variance (PERMANOVA) with the Adonis function in the R package, Vegan [15]. False discovery rate (FDR) was computed to adjust for multiple hypothesis testing by Benjamini-Hochberg. Pathway overrepresentation analysis was performed based on the hypergeometric test and relative betweenness centrality in the server, MetaboAnalyst 4.0 [16].

## 3. Results and Discussion

### 3.1. Integrative Metabolic Profiles of Rice Seeds Based on GC-TOF MS and LC-Orbitrap MS

Untargeted metabolic profiling was performed by gas chromatography time-of-flight mass spectrometry (GC-TOF MS) and liquid chromatography Orbitrap mass spectrometry (LC-Orbitrap MS). A total of 156 metabolites were acquired in the combined MS analysis. The identified compounds were classified by chemical taxonomy (HMDB, http://www.hmdb.ca) (accessed on 20 November 2020). Major subclasses included amino acids, peptides, and analogs (30%); carbohydrates and carbohydrate conjugates (17%); fatty acids and conjugates (7%); and purines and purine derivatives (4%) (Figure 1). The subclass amino acids-peptides-analogs was further categorized into L-alpha-amino acids, alpha-amino acids, alanine, and derivatives, while carbohydrate-carbohydrate conjugates consisted of O-glycosyl compounds, sugar alcohols, sugar acids, and derivatives. For statistical analysis, the datasets were combined following MS total useful signal (MSTUS) and were implemented in Normalization and Evaluation of Metabolomics Data (NOREVA), allowing comparisons to be made among datasets from different MS platforms.

First, the metabolomic phenotype was characterized based on unsupervised multivariate statistics (PCA). The score plot by PCA shows the unbiased separation of the profiles according to the polishing degree (brown vs. white) by component 1 (47.2%) and the variety (Odae vs. Ilpum) by component 2 (10.7%) (Figure 2A). The authors did not identify a region-dependent cultivation cluster except for partial discrimination within the brown rice of the Odae cultivar (blue circles). However, cultivar dominated the rice metabolic phenotype over potential geoenvironmental factors (different cultivation regions) which differed from other crops cultivated in Korea [5,12]. The percentage of metabolic variation explained by each factor (polishing type, variety, and cultivation region) was determined based on permutation multivariate analysis of variance (PERMANOVA) with the Adonis function. Consistently, the analysis confirmed that the highest level of the explained variance across the metabolome was due to the polishing process (37.5%), followed by the variety (13.5%), and cultivation region (4.6%) (Figure 2B).

Most compounds (127 out of 156 identified compounds) were more abundant in brown rice than in white rice. The highest fold change was detected for cafestol (PubChem CID: 108052, 12-fold difference) and reduced glutathione (PubChem CID: 124886, 10-fold difference). Cafestol is a natural diterpene that is commonly found in coffee beans. Diterpenes have been observed to have potential bioactivities, including anti-inflammatory activity, hepatoprotective effects, and antitumor capacity [17], which may be due to its increased levels of glutathione [17]. Thirteen compounds were found at 5- to 10-fold increases, and 34 metabolites ranged from 3- to 5-fold differences. Others (72 metabolites) were within the range of one- to two-fold changes (*p* < 0.05, Appendix A). In contrast, three compounds were found to be at significantly higher levels in white rice than in brown rice ((2-hydroxypyridine (PubChem CID: 34037), glycerophospho-*N*-palmitoyl ethanolamine (PubChem CID: 53393933), and lactic acid (PubChem CID: 612)). The compounds were found to be 1.2- to 1.8-fold richer in white rice compared to brown rice. The overall enriched metabolites in brown rice may provide molecular evidence for better nutritional quality, and thus, potential health benefits [18].

### 3.2. Unique Metabolic Signatures According to Variety Type (Odae vs. Ilpum)

Variety was inferred as the second dominant factor from PCA and PERMANOVA and was comparatively analyzed between Odae and Ilpum; based on Student’s *t*-test, univariate statistics showed overall higher contents in Odae for both white and brown rice (Appendix A). A previous study reported few differences in the physiological properties and genetic variation in the Odae cultivar bred for early-ripening traits in comparison to the Ilpum cultivar [6]. However, the metabolic profiles were substantially different between the two varieties. The Odae cultivar showed 82% and 72% of metabolites for brown and white rice, respectively, which was significantly higher than those found in the Ilpum cultivar. In particular, the highest fold enrichment was observed for trigonelline (PubChem CID: 5570) in the Odae cultivar (fold change of 7.4 and 5.0 for brown and white rice, respectively). Plant alkaloids, which are abundant in coffee, have been reported to have antidiabetic effects via hypoglycemic effects and the inhibition of intestinal glucose uptake. In addition, the compound has been suggested to have a neuroprotective effect by potential inhibitory activities, including γ-aminobutyric acid (GABA) receptor, acetylcholinesterase, and amyloid-β peptide aggregation [19].

Using pathway overrepresentation analysis, the metabolic profiles were further characterized. Despite the overall enrichment in the Odae cultivar, the white rice of the Ilpum cultivar showed exclusive enrichment of the metabolites involved in arginine biosynthesis (glutamine, PubChem CID: 5961; ornithine, PubChem CID: 6262) (Appendix A). Others were pyrimidine metabolism, steroid biosynthesis (squalene, PubChem CID: 638072; cholesterol, PubChem CID: 5997), and sesquiterpenoid-triterpenoid biosynthesis (squalene, PubChem CID: 638072). In addition, differential regulation of the same pathways was found for arginine-proline metabolism in which agmatine (PubChem CID: 199) and ornithine (PubChem CID: 6262) were higher in Ilpum, whereas proline (PubChem CID: 145742), putrescine (PubChem CID: 1045), and 4-guanidinobutanoate (PubChem CID: 500) were higher in Odae.

To exclusively delineate the metabolic differences derived from variety (Odae vs. Ilpum), OPLS-DA was applied to separate linearly related (joint) and unrelated (orthogonal) factors [20]. Indeed, the resultant score plot indicated that the polishing type-driven variance was projected to the *y*-axis (unrelated vector), and the variance originating from the variety was aligned with the *x*-axis (Figure 3A). The model showed that the highest levels of explained variance and predictability were R2Y and Q2 at 0.978 and 0.935, respectively, with an R^2^Y intercept value of 0.673 and a Q^2^Y intercept value of −0.811 (five-fold cross-validation).

The subsequent loading scatter plot, based on the OPLS-DA model, identified the metabolites with variety specificity (Figure 3B). The metabolites close to the group node (Odae) indicated a strong correlation with the Odae cultivar (yellow box). The metabolites with the highest specificity were myristic acid (PubChem CID: 11005), isoferulic acid (PubChem CID: 736186), sphingosine (PubChem CID: 5280335), 2-amino-1,3,4-octadecanetriol (PubChem CID: 122121), trigonelline (PubChem CID: 5570), and nicotinic acid (PubChem CID: 938) (Figure 3C). Others showed higher abundances in Odae than in Ilpum but with an interactive effect of the polishing degree. Likewise, the metabolites that were highly correlated with Ilpum (blue box) showed higher abundances, and the differences were amplified by the polishing type. The metabolites included squalene (PubChem CID: 638072), glutamine (PubChem CID: 5961), and oxalic acid (PubChem CID: 971) (Figure 3D). In addition, a global approach based on chemical structural similarity and ontology mapping identified a chemical class, namely indoles, that showed variety-dependent abundances. This relationship was particularly identified in white rice (Appendix A). Significant differences were found for 6-methylquinoline (PubChem CID: 7059) and skatole (PubChem CID: 6736) (*p* < 0.05) with the moderate enrichment of tryptamine (PubChem CID: 1150) (*p* = 0.06). As a reactant of indole-3-acetate, skatole is biotransformed by acetate carboxyl-lyase which is encoded in some microorganisms (e.g., the *Clostridium* genus). The different content of the metabolites implies a putative interaction between crop and microbiome. However, it is important to note that the indole derivative has been detected in some flowers as well.

A previous report [6] determined that there was no significant difference in the biochemical properties and DNA levels between the two varieties. Thus, comprehensive metabolomic profiling proved to be a powerful method to discriminately characterize the molecular traits and, ultimately, link them to physiology.

### 3.3. Metabolic Differences Associated with Geoenvironmental Factors (Cultivation Region)

Region-specific profiles were first evaluated based on OPLS-DA within predefined factors (white rice-Ilpum, white rice-Odae, brown rice-Ilpum, and brown-Odae). Overall, all four models indicated high levels of discrimination power (explained level) and practicability (Figure 4). To note, the region-specific features were successfully delineated despite the predominant factors (polishing type and cultivar). The brown rice of the Ilpum variety cultivated in the Sangju (SJ) and Chuncheon (CC) regions showed a close association which was discriminated from the other regions (Suwon (SW), Jeonju (JJ), and Cheongju (CJ)) (R2Y = 0.993, Q2 = 0.746) (Figure 4D). The metabolites of lyxitol (PubChem CID: 94154), butane-2,3-diol (PubChem CID: 262), and xylitol (PubChem CID: 6912) were the strongest contributors to the discriminant model. The white rice of the Ilpum cultivar was distinctively clustered along with each region in which the SJ and SW regions were separated from the CC, JJ, and CJ regions based on the first component (R2Y = 0.999, Q2 = 0.806) (Figure 4C). The metabolites with the top Variable Importance for the Projection (VIP) scores were 6-methylquinoline (PubChem CID: 7059) and tryptamine (PubChem CID: 1150) (VIP score > 1.6). Likewise, the Odae cultivar was regionally characterized by the relatedness between SJ and SW for both brown and white rice (Figure 4A,B). The major contributors were uric acid (PubChem CID: 1175), galactinol (PubChem CID: 11727586), and succinic acid (PubChem CID: 1110) for brown rice, whereas 2,5-dihydroxypyrazine (PubChem CID: 23368901), sorbitol (PubChem CID: 5780), and fructose (PubChem CID: 2723872) were the metabolites with the top VIP score for white rice.

To further investigate the cultivation of region-dependent metabolic features, the authors normalized the metabolite contents by autoscaling and summing them within each chemical class. In general, the chemical, class-wise metabolic features were coordinated with score plots computed by the OPLS-DA models.

### 3.4. Odae Cultivar Characteristics Related to Cultivation Region

Overall, the profiles of CC and JJ showed relatively enriched contents compared to other regions. The white rice cultivated in JJ showed the highest levels of amino acids (red bar) and carbohydrates (orange bar) in contrast to the SJ region (Figure 5B). The CC region showed moderately higher levels of all chemical classes, whereas the CJ regions presented lower levels except for carbohydrates. The SW region featured the highest level of fatty acyls and the lowest level of carbohydrates. For brown rice, the profiles of the CC region showed the highest contents in all classified groups. Likewise, the brown rice of CJ and CC showed relatively higher contents. Three chemical classes (amino acids, carbohydrates, and fatty acyls) were at the highest levels in the rice cultivated in the CC region. Similarly, the higher contents of amino acids and carbohydrates were characteristic of the CJ and JJ regions, whereas the SJ and SW rice presented the lowest abundances, particularly of amino acids and carbohydrates (Figure 5C).

*Ilpum cultivar characteristics related to cultivation region.* For white rice, the Ilpum variety cultivated in SJ showed the highest contents of amino acids, carbohydrates, and purines (Figure 5D). The profiles of the SW region were at moderately high levels compared to the other regions, while the rice in CJ was characterized by the lowest contents of all four classes. Two regions, CC and JJ, showed similar patterns in which fatty acyls were at moderately high levels, and the three other classes were at lower contents. The overall enriched contents were determined in the brown rice of the SJ and CC regions. The rice of the JJ region featured the lowest contents of amino acids, carbohydrates, and purines, while all chemical classes showed moderate levels in the profiles of CJ and SW (Figure 5E).

Finally, the putative linkage between metabolic features and geographic traits was investigated. Despite the minor geoenvironmental influence on the rice metabolic profiles, a partial association was observed, particularly in the brown rice of the Odae cultivar. This type of rice showed distinctive clusters by PCA (Figure 2). The subsequent score plot analysis explicated the distinctions between the two areas (Appendix A). Among the clustered regions (CC, CJ, and JJ), a comparable pattern of annual rainfall was observed in the CC and JJ regions (Figure 5F). Moreover, the soil texture was similar among the three regions (Figure 5G). The regions’ textural class was justified by the relatively higher portion of loam. The high dependency of metabolite contents on soil type has been reported in soybeans cultivated in Korea [5].

## 4. Conclusions

The main goal of the current study was to systematically deconvolute the complicated layers of rice metabolic profiles that were overlaid by multiple factors (polishing degree, variety, and cultivation region). Our results revealed the relative contribution of these factors to the metabolic profiles of rice cultivated in five Korean regions. Unlike our previous investigation of soybean and sesame seeds, rice metabolic profiles were more strongly influenced by variety (early- or late-ripening cultivar) rather than cultivation region, and all followed polishing degree (brown or white rice). Although we do not propose a comprehensive linkage between the region-specific metabolic profiles and geo-environmental factors, the unique metabolic signatures were clearly identified according to five cultivation areas. Although the five cultivation areas varied, the unique metabolic signatures were not exclusively determined by the other components (e.g., polishing degree and variety).

Utilizing comprehensive genotyping under well-defined environments, the current study’s outcomes provide greater detail and mechanistic insight into the biochemical consequences and metabolic physiology of rice that are complicated by multiple factors in a crop-specific manner, including microbiome–plant interactions. In addition, this approach can meet the growing public demand for the authentic discrimination of the variety and cultivation region of agricultural products.

## Figures and Tables

**Figure 1 foods-10-00711-f001:**
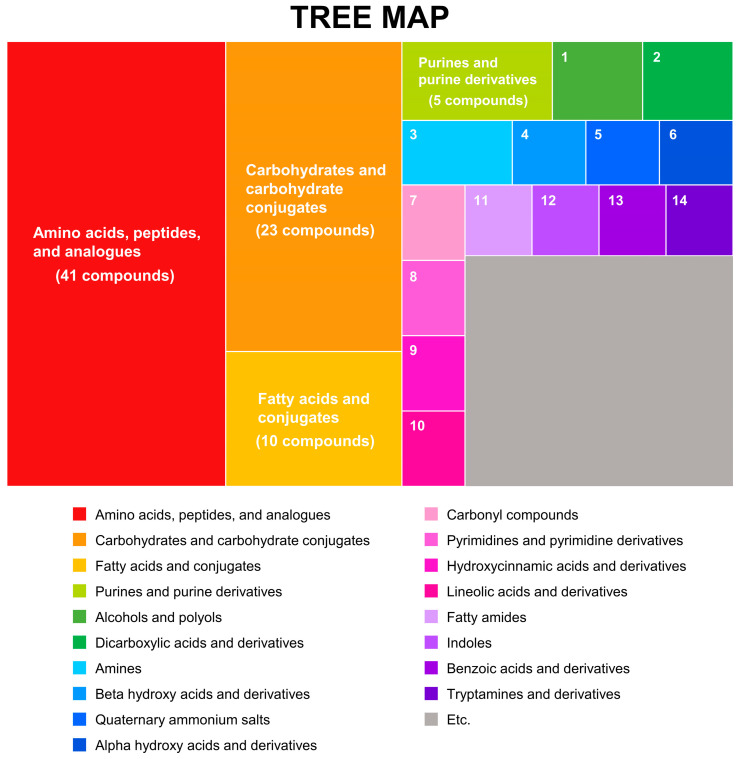
Chemical classification of rice metabolic profiles based on chemical taxonomy. The classification was conducted by chemical taxonomy from Human Metabolome Database (HMDB) (http://www.hmdb.ca) (accessed on 11 July 2020).

**Figure 2 foods-10-00711-f002:**
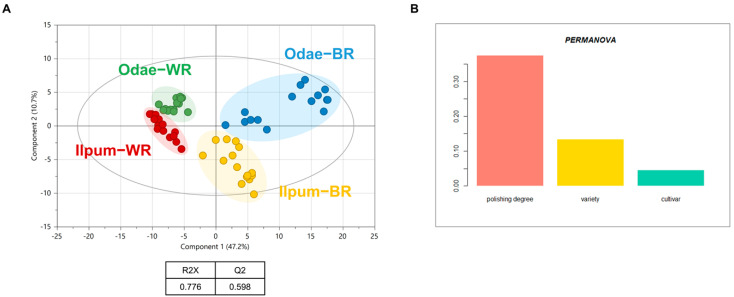
Metabolic phenotypes of the rice profiles using multivariate statistical analysis. (**A**) The score plot of the rice profiles based on principal component analysis (PCA). The variance is best explained by the t1 vector (component 1, 47.2%) which separated the profiles according to polishing degree (white vs. brown rice). The cultivar (Odae vs. Ilpum) were separated by the t2 vector (component 2, 10.7%). White rice (WR) and Brown rice (BR) indicate white rice and brown rice, respectively. (**B**) The explained levels of total variation in the metabolic profiles of rice explained by the major factors (polishing type, variety, and cultivation region) computed based on Permutational multivariate analysis of variance (PERMANOVA).

**Figure 3 foods-10-00711-f003:**
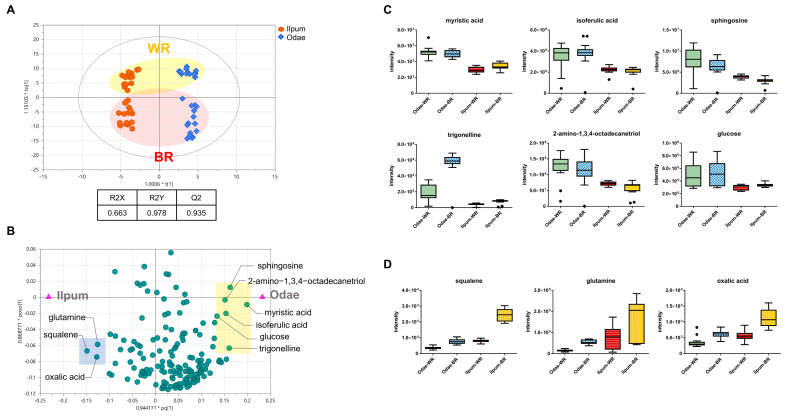
Cultivar-dependent metabolite profiles based on orthogonal projection to latent structures-discriminant analysis (OPLS-DA). (**A**) The score plot of the rice profiles and (**B**) loading scatter plot based on OPLS-DA. The metabolites in the yellow box present a high correlation with the Odae cultivar, whereas those in the blue box show a high correlation with the Ilpum cultivar. Box and whisker plots of the metabolites in the yellow box (**C**) and the blue box. (**D**) WR and BR indicate white rice and brown rice, respectively.

**Figure 4 foods-10-00711-f004:**
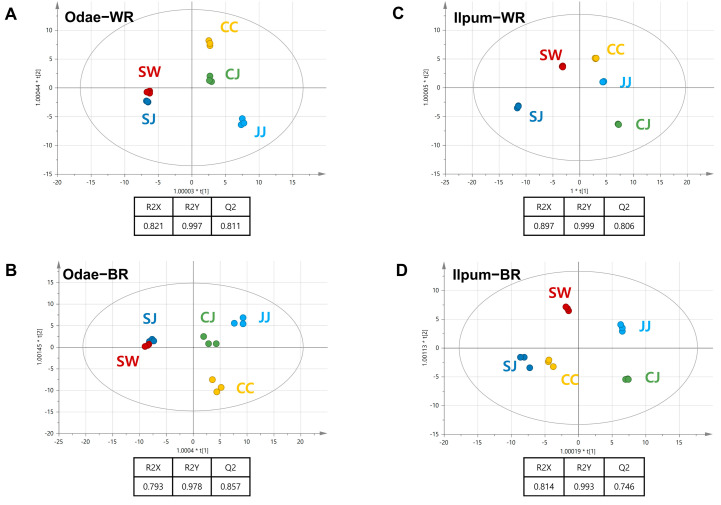
Cultivation-regional specificity of the rice profiles based on orthogonal projection to latent structures-discriminant analysis (OPLS-DA). The score plot of the rice profiles cultivated in 5 different regions for Odae-white rice, (**A**) for Odae-brown rice, (**B**) for Ilpum-white rice, and (**C**) for Ilpum-brown rice. (**D**) CJ: Cheongju (Chungcheongbuk-do), SJ: Sangju (Gyeongsangbuk-do), SW: Suwon (Gyeonggi-do), CC: Cuncheon (Gangwon-do), JJ: Jeonju (Jeollabuk-do).

**Figure 5 foods-10-00711-f005:**
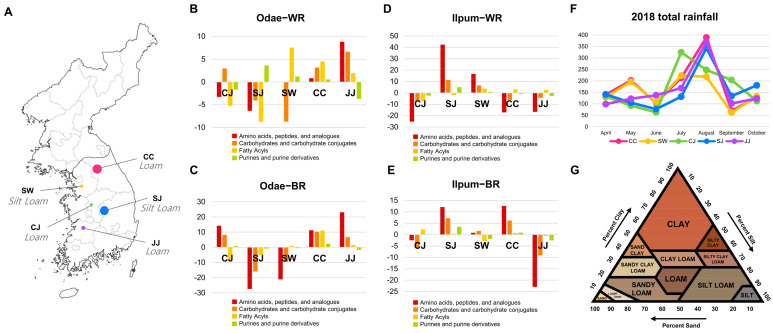
The metabolic signatures with regional specificity. The metabolites were classified into 4 major classes and normalized by autoscaling. The *y*-axis is the sum of the normalized intensities of the metabolites in each class. (**A**) Geographic map of the cultivation regions in Korea. The relative abundances of 4 chemical classes in white rice and (**B**) brown rice (**C**) of the Odae cultivar and white rice and (**D**) brown rice (**E**) of the Ilpum cultivar. (**F**) Total rainfall (from April to October 2018) and (**G**) soil texture of 5 different cultivation regions. Information on the soil texture was acquired from the National Institute of Agricultural Sciences (http://soil.rda.go.kr/soil/index.jsp) (accessed on 20 June 2020), and the triangle chart is presented based on information from the United States Department of Agriculture (USDA). CJ: Cheongju (Chungcheongbuk−do), SJ: Sangju (Gyeongsangbuk−do), SW: Suwon (Gyeonggi−do), CC: Cuncheon (Gangwon−do), JJ: Jeonju (Jeollabuk−do).

## Data Availability

The data presented in this study are available on request from the corresponding author.

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
