# Peer review of "Unique Metabolic Profiles of Korean Rice According to Polishing Degree, Variety, and Geo-Environmental Factors"

_foods, 2021, doi:10.3390/foods10040711_

Round 1

Reviewer 1 Report

This article concerns with a study of the effects of polish, genotype and environmental factors and their interactions on the metabolic profile of two Korean rice varieties. The results are detailed, well explained and, as far as I can determine, the work was conducted in a meticulous and thorough way.

Regarding the quality of the manuscript, some revisions are needed:

-English should be revised in the introduction section, that could also be improved.

-In the introduction no reference to results have to be included.

-The use of personal pronouns is not recommended.

-In the abstract and conclusion sections the subject of tryptophan metabolites involved in plant-microbe interaction is mentioned, but in the text any reference on the topic is included (only nutritional properties of metabolites are mentioned.

-Table S1 could be shorted, being the regions the same for the two cultivars it is redundant to repeat the name 2 times.

-Table S2 explain what is fold change and FDR (also in table S3)

-The caption of Figure S2 should be more clear…maybe higher in Odae than Pilmun?

Author Response

Comments and Suggestions for Authors

This article concerns with a study of the effects of polish, genotype and environmental factors and their interactions on the metabolic profile of two Korean rice varieties. The results are detailed, well explained and, as far as I can determine, the work was conducted in a meticulous and thorough way.

Response: We appreciate very careful and insightful comments on the manuscript. Accordingly, we have revised the manuscript and reflected all comments in the revised manuscript.

Regarding the quality of the manuscript, some revisions are needed:

- English should be revised in the introduction section, that could also be improved.

Response: According to the Reviewer’s comment, we have revised manuscript.

- In the introduction no reference to results have to be included.

Response: We appreciate the Reviewers comments. We have edited the introduction part

- The use of personal pronouns is not recommended.

Response: We agree with the Review’s comment. Thus, we have revised the manuscript in a way personal pronouns usage is minimized.

-In the abstract and conclusion sections the subject of tryptophan metabolites involved in plant-microbe interaction is mentioned, but in the text any reference on the topic is included (only nutritional properties of metabolites are mentioned.

Response: Thanks to the Reviewer’s comment, we have detailed the finding with related references in the revised manuscript.

-Table S1 could be shorted, being the regions the same for the two cultivars it is redundant to repeat the name 2 times.

Response: According to the Reviewer’s suggestion, we have edited the table in the current manuscript.

-Table S2 explain what is fold change and FDR (also in table S3)

Response: We have clarified them in the revised manuscript.

-The caption of Figure S2 should be more clear…maybe higher in Odae than Ilpum?

Response: Thank you for the point. We have revised it in the revised manuscript.

Reviewer 2 Report

Summary

This article presents a metabolite analysis of diverse types (brown and white) and cultivars of rice (Ilpum vs Odae), as well as the variation in different cultivation regions. the methods are adequate and the statistical treatment is correct.

General comment

The article is interesting and contain valuable information based in original data. Nevertheless it has some drawbacks in the presentation that need to be corrected. Please pay attention at the corrections marked in yellow in the PDF annex.

Comments by sections

Introduction:

“For rice and bran, the residue after the milling process (during the transition of brown rice to white rice) contains various types of beneficial nutrients and is thus a major factor in determining nutritional value. Others include genotype (variety) …”

This is not correct, because the residue is genotype-dependent (and variety-dependent).

Materials and Methods:

The information is summarized in the supplementary information (Table S1).

But I could not see Table 1 in the supplementary information. Please add this table to the supplementary information.

Results:

Lines 694-695:

A previous study reported few differences in the physiological properties and genetic variation in the Odae cultivar bred for early-ripening traits from the Ilpum cultivar.

Please give a reference for the reported study

Figure 4b in the body of the Figure Correct Odea to Odae

Please explain in Figure 5G …how can the reader get information for the soil texture in each of the five localities, if the figure legend says the following?:

(G) soil texture of 5 different cultivation regions. Information on the soil texture was acquired from 1094 the National Institute of Agricultural Sciences (http://soil.rda.go.kr/soil/index.jsp), and the triangle chart is presented 1095 based on information from the United States Department of Agriculture (USDA).?

Minor changes suggested:

Line 665 adonis with uppercase Adonis

Change:

the size and number of soybean seeds were markedly decreased

To:

the size and number of soybean seeds markedly decreased

Author Response

Comments and Suggestions for Authors

Summary

This article presents a metabolite analysis of diverse types (brown and white) and cultivars of rice (Ilpum vs Odae), as well as the variation in different cultivation regions. the methods are adequate and the statistical treatment is correct.

General comment

The article is interesting and contain valuable information based in original data. Nevertheless, it has some drawbacks in the presentation that need to be corrected. Please pay attention at the corrections marked in yellow in the PDF annex.

Response: We thank careful comments and insightful suggestion on the manuscript. Accordingly, we have revised the manuscript and reflected all comments in the revised manuscript.

Comments by sections

Introduction:

“For rice and bran, the residue after the milling process (during the transition of brown rice to white rice) contains various types of beneficial nutrients and is thus a major factor in determining nutritional value. Others include genotype (variety) …”

This is not correct, because the residue is genotype-dependent (and variety-dependent).

Response: We agree with the Reviewer’s comment. We meant bran and germ, which is removed for white rice during polishing process, is nutrition-rich part. Now we have clarified it in the revised manuscript.

Materials and Methods:

The information is summarized in the supplementary information (Table S1).

But I could not see Table 1 in the supplementary information. Please add this table to the supplementary information.

Response: Thank you for the check. We have included Table S1 in the revised manuscript.

Results:

Lines 694-695:

A previous study reported few differences in the physiological properties and genetic variation in the Odae cultivar bred for early-ripening traits from the Ilpum cultivar.

Please give a reference for the reported study

Response: We appreciate the careful check. Accordingly, we have included a reference related to the statement.

Figure 4b in the body of the Figure Correct Odea to Odae

Response: We have now fixed it. Thank you.

Please explain in Figure 5G …how can the reader get information for the soil texture in each of the five localities, if the figure legend says the following?:

(G) soil texture of 5 different cultivation regions. Information on the soil texture was acquired from the National Institute of Agricultural Sciences (http://soil.rda.go.kr/soil/index.jsp), and the triangle chart is presented based on information from the United States Department of Agriculture (USDA).?

Response: We agree with the Reviewer’s opinion. We have now clarified the figure and the legend in the current manuscript.

Minor changes suggested:

Line 665 adonis with uppercase Adonis

Change:

the size and number of soybean seeds were markedly decreased

To:

the size and number of soybean seeds markedly decreased

Response: Thanks to the Reviewer’s careful check, we have corrected them in the revised manuscript.